# Classification performance of administrative coding data for detection of invasive fungal infection in paediatric cancer patients

Jake C. Valentine[1,2,3]*, Leon J. Worth[1,3,4], Karin M. Verspoor[1,5], Lisa Hall[1,6], Daniel K. Yeoh[1,7], Karin A. Thursky[1,3,4], Julia E. Clark[8], Gabrielle M. Haeusler[1,2,3,4,9]

1 National Centre for Infections in Cancer, Peter MacCallum Cancer Centre, Melbourne, Victoria, Australia, 2 Paediatric Integrated Cancer Service, Royal Children's Hospital, Parkville, Victoria, Australia, 3 Sir Peter MacCallum Department of Oncology, University of Melbourne, Parkville, Victoria, Australia, 4 Department of Infectious Diseases, Peter MacCallum Cancer Centre, Melbourne, Victoria, Australia, 5 School of Computing and Information Systems, University of Melbourne, Parkville, Victoria, Australia, 6 School of Public Health, University of Queensland, Brisbane, Queensland, Australia, 7 Department of Infectious Diseases, Perth Children's Hospital, Perth, Western Australia, Australia, 8 Infection Management Service, Queensland Children's Hospital, Brisbane, Queensland, Australia, 9 Infectious Diseases Unit, Department of General Medicine, Royal Children's Hospital, Parkville, Victoria, Australia

* Jake.Valentine@petermac.org

**Data Availability Statement:** Data sharing is restricted to the researchers, clinical staff and the

## Abstract

### Background

Invasive fungal infection (IFI) detection requires application of complex case definitions by trained staff. Administrative coding data (ICD-10-AM) may provide a simplified method for IFI surveillance, but accuracy of case ascertainment in children with cancer is unknown.

### Objective

To determine the classification performance of ICD-10-AM codes for detecting IFI using a gold-standard dataset (r-TERIFIC) of confirmed IFIs in paediatric cancer patients at a quaternary referral centre (Royal Children's Hospital) in Victoria, Australia from 1st April 2004 to 31st December 2013.

### Methods

ICD-10-AM codes denoting IFI in paediatric patients (<18-years) with haematologic or solid tumour malignancies were extracted from the Victorian Admitted Episodes Dataset and linked to the r-TERIFIC dataset. Sensitivity, positive predictive value (PPV) and the $F_1$ scores of the ICD-10-AM codes were calculated.

### Results

Of 1,671 evaluable patients, 113 (6.76%) had confirmed IFI diagnoses according to gold-standard criteria, while 114 (6.82%) cases were identified using the codes. Of the clinical IFI cases, 68 were in receipt of ≥1 ICD-10-AM code(s) for IFI, corresponding to an overall sensitivity, PPV and $F_1$ score of 60%, respectively. Sensitivity was highest for proven IFI (77%

individual patient's healthcare provider participating in the project analysis. The ethics approval (HREC/59636/RCHM-2019) maintains that public data provision is contingent on authorisation from the Principal Investigator and associated ethics applications to the Royal Children's Hospital Human Research Ethics Committee for researchers who meet the criteria for access to confidential data. Contact details for the relevant ethics committee to which data requests may be sent is rch.ethics@rch.org.au.

**Funding:** J.C.V. was supported by an Australian Government Research Training Program Scholarship (grant number: 290465) awarded by the University of Melbourne (URL: https://www.education.gov.au/research-training-program) and a Cardinal Health Infection Control Scholarship awarded by the Australasian College for Infection Prevention and Control (URL: https://www.cardinalhealth.com/en.html). The TERIFIC study was supported by an Investigator Initiated Grant (grant number: IN-AU-131-1314) from Gilead Sciences, Inc. (https://www.gilead.com/). The funders had no role in study design, data collection and analysis, decision to publish, preparation or review of the manuscript.

**Competing interests:** I have read the journal's policy and the authors of this manuscript have the following competing interests: G.M.H. and J.E.C. report investigator-initiated grant funding from Gilead Sciences, Inc. K.M.V. reports grant funding from the National Health and Medical Research Council. All other authors declare no conflicts of interest relevant to this article. This does not alter our adherence to PLOS ONE policies on sharing data and materials.

[95% CI: 58–90]; $F_1$ = 47%) and invasive candidiasis (83% [95% CI: 61–95]; $F_1$ = 76%) and lowest for other/unspecified IFI (20% [95% CI: 5.05–72%]; $F_1$ = 5.00%). The most frequent misclassification was coding of invasive aspergillosis as invasive candidiasis.

## Conclusion

ICD-10-AM codes demonstrate moderate sensitivity and PPV to detect IFI in children with cancer. However, specific subsets of proven IFI and invasive candidiasis (codes B37.x) are more accurately coded.

## Introduction

Invasive fungal infections (IFIs) represent significant challenges in the management of paediatric cancer patients with impaired immunity [1–3] and are an important cause of morbidity and mortality [1, 4]. Current methods for detecting IFI are manual, time consuming and often labour intensive [1, 2, 5, 6], and are reliant on a suite of clinical, laboratory and radiological data. There is therefore limited capacity to routinely capture IFIs to assess the epidemiology, detect potential outbreaks and inform optimal antifungal use in children with cancer [7].

Uniform case definitions for IFI are widely accepted as measurable outcomes in clinical trials (i.e. European Organization for Research and Treatment of Cancer/Invasive Fungal Infections Cooperative Group and the National Institute of Allergy and Infectious Diseases Mycoses Study Group [EORTC/MSG]) [8]. However, these are complex and require detailed case review. Administrative coding data possess potentially favourable attributes for simplified surveillance [9], including standardised classification and availability of specific codes for yeast and mould infections [4, 10]. In Australia, the *International Statistical Classification of Diseases and Related Health Problems*, *Tenth Revision*, *Australian Modification* (ICD-10-AM) are a monohierarchical, codified, medical lexicon used for coding inpatient diagnoses and is a commonly used ontology to inform activity-based funding models [11].

Earlier data have suggested the sensitivity of administrative coding data for classifying invasive aspergillosis to be moderate (63%) [12], but findings were restricted to filamentous fungi in adult allogeneic and autologous haematopoietic stem cell transplantation recipients and excluded invasive candidiasis, one of the most prevalent IFIs in the paediatric haematology-oncology setting [1]. Despite the high incidence and poor survival prognoses of IFI in cancer patients [13], there is a paucity of available evidence examining the utility of administrative coding data for reliable and reproducible surveillance of IFI in vulnerable paediatric cancer populations.

The objectives of this study were to: (i) determine the sensitivity, positive predictive value (PPV) and $F_1$ score of administrative coding data for case ascertainment of IFI; and (ii) describe the misclassification rate of ICD-10-AM in paediatric haematology-oncology patients.

## Materials and methods

### Study design and population

This was a retrospective, single-site, cohort study of paediatric patients (<18-years) diagnosed with a haematological malignancy or solid tumour neoplasm between the 1st April 2004 and 31st December 2013 at the Royal Children's Hospital (RCH) in Melbourne, Victoria, Australia. Study design was consistent with criteria endorsed in the STrengthening the Reporting of OBservational

studies in Epidemiology (STROBE; S1 Table) [14] and the REporting of studies Conducted using Observational Routinely-collected health Data (RECORD; S2 Table) statements [15].

## Gold-standard invasive fungal infection dataset

Data collected as part of the multisite The Epidemiology and Risk Factors for Invasive Fungal Infections in Immunocompromised Children (TERIFIC) study and restricted to episodes collected at the RCH (denoted as r-TERIFIC), were used for the current study [1, 2]. Detailed study methodology is available elsewhere [1, 2]. Briefly, this 10-year retrospective study identified all episodes of IFI in children with cancer or haematological malignancy from hospital microbiology, pharmacy-dispensing, radiology, oncology diagnostic and clinical management records as well as diagnostic coding data. Invasive fungal infection episodes were classified as proven, probable, possible or modified possible in accordance with EORTC/MSG criteria [8] and modifications described elsewhere [1, 2].

## Administrative coding dataset

Episode-level, administrative coding data were sourced from the Victorian Admitted Episodes Dataset (VAED) and mapped to each patient record captured in the r-TERIFIC dataset. The VAED is Australia's largest hospital morbidity database, and consists of diagnostic ICD-10-AM and procedural *Australian Classification of Health Interventions* (ACHI) codes for paediatric cancer patients admitted to private and public hospitals in Victoria [16].

Patients with haematological malignancy or a solid tumour were defined using the principal diagnosis codes denoting a primary malignant neoplasm (ICD-10-AM codes: C00.x - C76.x, C80.x, C81.x [0/1]—C88.x [0/1], C90.x [0/1]—C96.x [0/1], and D46), where "x" denotes any number (S3 Table). Invasive fungal infection was defined when an additional diagnosis code (Australian Coding Standards 0002 *Additional diagnoses* [17]) denoting IFI was reported in the VAED (ICD-10-AM codes: B37.x, B42.x - B50.x) (S3 Table). Hospitalisations for autologous or allogeneic haematopoietic stem cell transplantation were defined by corresponding ACHI codes 13706–00, -06, -07, -08, -09, -10 [802] (S4 Table). Updates to the ICD-10-AM and ACHI codes from the Third to Eighth Edition were elucidated. Duplicate IFI codes denoting the same IFI in the same hospitalisation, as well as those reported at the time of admission in subsequent hospitalisations, were considered the same IFI and were counted only once per patient. Multiple discrete IFI codes appearing in the same hospitalisation per patient were counted as separate IFI episodes. Accordingly, patients with ≥2 mutually exclusive gold-standard IFI diagnoses in the r-TERIFIC dataset, diagnosed in the same or in discrete hospitalisations, were counted as individual gold-standard cases for each IFI diagnosis (for example, one patient with both invasive aspergillosis and invasive candidiasis was counted as one case of invasive aspergillosis and one case of invasive candidiasis). Index hospitalisation was defined as the first admission date at the RCH.

## Exclusion criteria

Cancer patients with superficial fungal infections (codes B36.x), including dermatophytes (codes B35.x), and patients with no underlying malignancy were excluded.

## Statistical analyses

For patient and clinical characteristic data, normality was tested on histogram analysis and the skewness and kurtosis test [18]. The mean (±standard deviation) and median (interquartile range) were reported for parametric and non-parametrically distributed data, respectively.

**Classification accuracy.** To determine the accuracy of ICD-10-AM codes for IFI case detection, sensitivity, PPV and $F_1$ scores were calculated, stratified by IFI type, EORTC/MSG classification and underlying cancer diagnosis [19, 20]. Sensitivity and PPV of the coding data were calculated as the number of clinically-confirmed IFI patients in receipt of at least one IFI code (i.e. true positives; cases where the ICD-10-AM code agrees with the clinical label) divided by the total number of clinically-confirmed IFI cases in the r-TERIFIC dataset and the total number of patients assigned an ICD-10-AM code for IFI (code positives), respectively. Exact binomial 95% confidence intervals (CI) were calculated for all sensitivity and PPV calculations. The $F_1$ score was used to measure the harmonic mean of the sensitivity and PPV of the coding data according to the formula [21]:

$$F_1 = \left( \frac{PPV^{-1} + sensitivity^{-1}}{2} \right)^{-1} = 2 \left( \frac{sensitivity \times PPV}{sensitivity + PPV} \right) \tag{1}$$

where $F_1$ ranges in $[0,1] = \{F_1 : 0 \leq F_1 \leq 1\}$ and an $F_1 = 1$ indicates perfect sensitivity and PPV. To identify which coding abstraction yields the highest sensitivity, PPV and $F_1$ score within each combination of IFI codes, the union of different ICD-10-AM code sets for IFI (represented as $A_k$) was evaluated. The union of code sets $A_1$ and $A_2$, denoted $A_1 \cup A_2$, is equivalent to the set of patients in the r-TERIFIC dataset that are correctly assigned either code $A_1$ ($Pr(A_1)$) or code $A_2$ ($Pr(A_2)$) or codes $A_1$ and $A_2$ ($Pr(A_1 \cap A_2)$). Classification performance was determined according to increasing numbers of assigned code sets ($Pr(A_1 \cup A_2 \cup \ldots \cup A_k)$). Sensitivity, PPV and $F_1$ estimates of 0% indicate IFI code sets that were *not* assigned to true positive cases in the r-TERIFIC dataset, denoted $A_1' \cap A_2'$. The number of different combinations ($C$) of codes ($n$) in increasing set sizes ($r$) was determined according to the following formula:

$$C_r^n = \frac{n!}{r!(n-r)!} = \binom{n}{r} \tag{2}$$

Classification statistics are reported in accordance with the Standards for Reporting Diagnostic (STARD) accuracy studies statement [22] (S5 Table).

**Misclassification rate.** Misclassification rate was calculated as a proportion of discordant-coded IFIs (e.g. the proportion of invasive candidiasis cases coded as invasive aspergillosis).

All statistical analyses were undertaken using Stata/SE v15.1 software (StataCorp® LLC, College Station, Texas, U.S.A.) A two-sided $p$ value $<0.05$ was considered statistically significant.

## Ethics

Ethics approval was granted by the Royal Children's Hospital Human Research Ethics Committee (project number: 59636) and the need for informed consent was waived in accordance with the National Statement on Ethical Conduct in Human Research 2007 (Updated May 2015) [23].

## Results

### Study population

From 1st April 2004 to 31st December 2013, there were 1,671 paediatric cancer patients admitted to RCH according to the coding dataset (Fig 1). Of the 1,671 cancer patients, 114 (6.82%) were in receipt $\geq$1 ICD-10-AM code denoting IFI in the coding dataset and 113 (6.76%) fulfilled gold-standard definitions for IFI in the r-TERIFIC dataset. Sixty-eight of the 113 patients (60%) in the r-TERIFIC dataset were coded with $\geq$1 IFI (Fig 1; Table 1). Of the 45 patients in

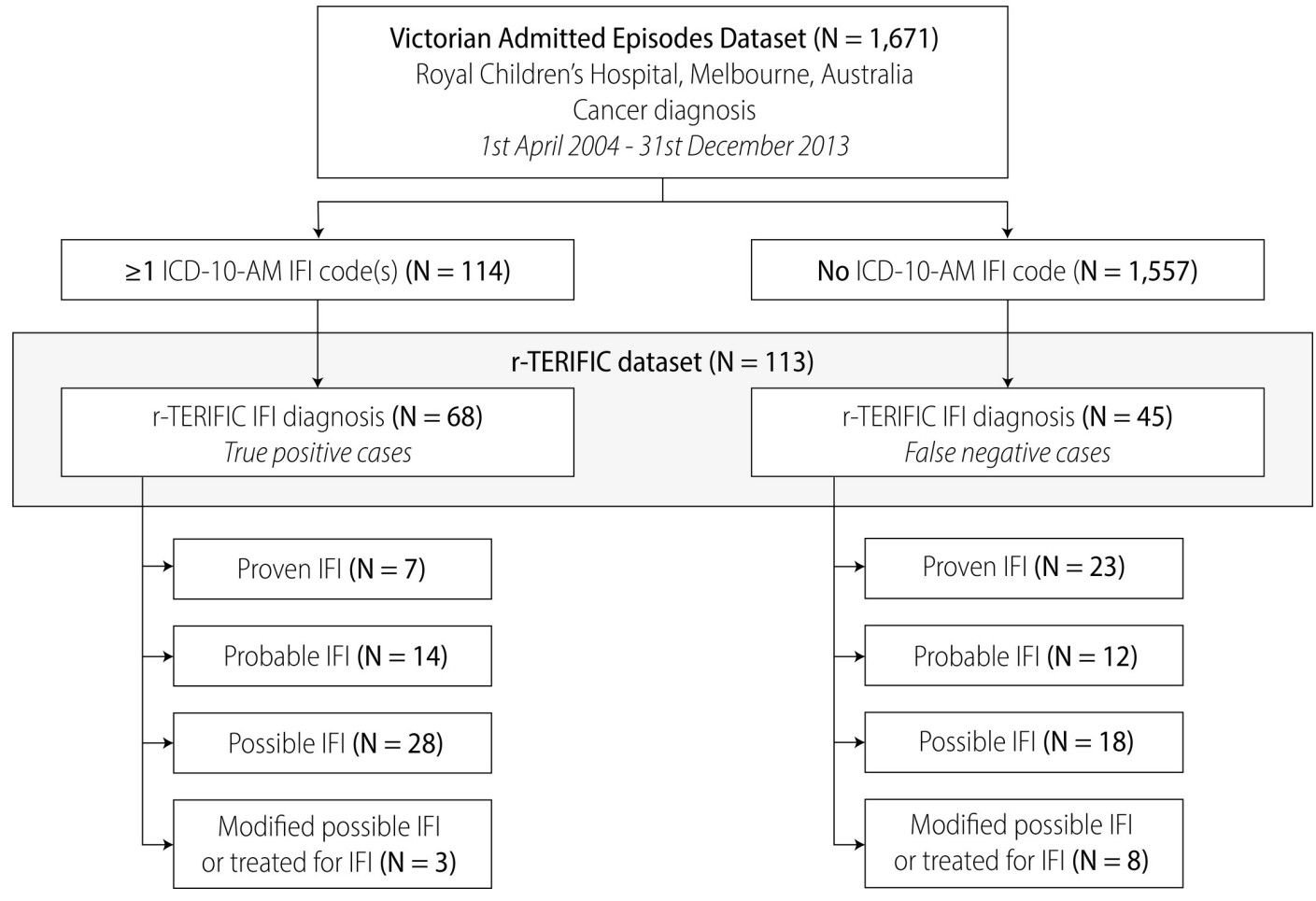

**Fig 1. Consort diagram of the study methodology and the number (*N*) of linked patient records across the r-TERIFIC and administrative coding datasets.**
ICD-10-AM, *International Statistical Classification of Diseases and Related Health Problems, Tenth Revision, Australian Modification*; IFI, invasive fungal infection; r-TERIFIC, The Epidemiology and Risk Factors for Invasive Fungal Infections in Immunocompromised Children (Royal Children's Hospital); VAED, Victorian Admitted Episodes Dataset.

the r-TERIFIC dataset that did not receive an ICD-10-AM code for IFI, nine had invasive aspergillosis (20%), 11 invasive candidiasis (24%) and 4 other/unspecified IFI (8.89%). There were 46 false positive cases in the coding dataset that were not captured in the r-TERIFIC

**Table 1. Clinical agreement between the gold-standard and administratively-coded cases of invasive fungal infection in the study cohort.**

| Administratively-coded IFI cases (VAED) | Gold-standard clinical IFI status (r-TERIFIC dataset) | | |
| --- | --- | --- | --- |
| | Positive | Negative | Total |
| **Positive** | *True positive* | *False positive* | 114 |
| | 68 | 46 | |
| **Negative** | *False negative* | *True negative* | 1,557 |
| | 45 | 1,512 | |
| **Total** | 113 | 1,558 | 1,671 |

Abbreviations: r-TERIFIC, The Epidemiology and Risk Factors for Invasive Fungal Infections in Immunocompromised Children (Royal Children's Hospital); VAED, Victorian Admitted Episodes Dataset.

**Table 2. Baseline characteristics of the study cohort, N = 1,671.**

| Patient and clinical characteristic | n (%) |
| --- | --- |
| Age (years; mean [± standard deviation]) | 7.54 [± 5.29] |
| Gender (male) | 926 (55) |
| Inpatient length of stay (days; median [IQR]) | 43 [12–93] |
| Admission to ICU | 63 (3.77) |
| Haematopoietic stem cell transplantation | |
| Autologous | 156 (9.34) |
| Allogeneic | 123 (7.36) |
| **Underlying malignancy** | |
| Haematological (N = 899) | |
| Acute lymphoblastic leukaemia [a] | 516 (31) |
| Acute myeloid leukaemia | 110 (6.58) |
| Non-Hodgkin lymphoma | 103 (6.16) |
| Hodgkin lymphoma | 92 (5.51) |
| Other | 78 (4.67) |
| Solid tumour (N = 772) | |
| Neuroblastoma | 397 (24) |
| Other | 375 (22) |

Abbreviations: ICU, intensive care unit; IQR, inter-quartile range.

[a] Including B- and T-cell variants.

dataset (Table 1), of which 38 (83%) were coded as 'candidiasis of other sites' (ICD-10-AM code: B37.88).

Baseline characteristics of the study cohort are presented in Table 2. Acute lymphoblastic leukaemia was the predominant underlying malignancy (n = 516; 31%), followed by neuroblastoma (n = 397; 24%) and acute myeloid leukaemia (n = 110; 6.58%) (Table 2). Of the 113 patients defined according to EORTC/MSG criteria, 46 (41%) were classified as possible, 30 (27%) proven, 26 probable (23%) and 11 (9.73%) modified possible or treated for IFI.

## Sensitivity, positive predictive value and F1 scores of the ICD-10-AM codes

Sensitivity, PPV and $F_1$ scores of the ICD-10-AM codes are provided in Table 3. Sixty-eight of the 113 IFI patients in the r-TERIFIC dataset were in receipt of ≥1 ICD-10-AM code(s) for IFI, resulting in an overall sensitivity of 60% (95% CI: 51–69; 68/113 cases) (Table 3). Sixty-eight of the 114 IFI-coded patients in the ICD-10-AM coding dataset were identified in the r-TERIFIC dataset (PPV: 60% [95% CI: 50–69]), resulting in an $F_1$ score of 60% (Table 3). After stratifying by type of IFI, invasive candidiasis codes resulted in the highest sensitivity, PPV and $F_1$ score of 83% (95% CI: 61–95), 70% (95% CI: 58–81) and 76%, respectively, followed by invasive aspergillosis (sensitivity: 42% [95% CI: 20–67]; PPV: 32% [95% CI: 15–54]; $F_1 = 36\%$) (Table 3). After stratifying by underlying malignancy, sensitivity for all coded IFI was highest in patients with neuroblastoma (88% [95% CI: 47–99]; $F_1 = 52\%$) and acute lymphoblastic leukaemia (69% [95% CI: 55–80]; $F_1 = 68\%$) (Table 3), and PPV was highest in patients with acute myeloid leukaemia (76% [95% CI: 55–91]; $F_1 = 66\%$) (Table 3). Overall classification performance of the IFI codes was highest for proven EORTC/MSG (sensitivity, 77% [95% CI: 58–90]; PPV, 34% [95% CI: 23–46]; $F_1 = 47\%$) and possible EORTC/MSG IFI diagnoses (sensitivity, 61% [95% CI: 45–75]; PPV, 41% [95% CI: 29–54]; $F_1 = 49\%$) in the studied population (N = 1,671).

**Table 3. Performance (in percent) of administrative coding data for case detection of proven, probable and possible invasive fungal infection.**

| Invasive fungal infection | All cancers (N = 1,671) | | | Acute lymphoblastic leukaemia (N = 516) | | | Acute myeloid leukaemia (N = 110) | | | Neuroblastoma (N = 397) | | |
|---|---|---|---|---|---|---|---|---|---|---|---|---|
| | Sensitivity [95% CI] | Positive predictive value [95% CI] | $F_1$ | Sensitivity [95% CI] | Positive predictive value [95% CI] | $F_1$ | Sensitivity [95% CI] | Positive predictive value [95% CI] | $F_1$ | Sensitivity [95% CI] | Positive predictive value [95% CI] | $F_1$ |
| **All EORTC/MSG (N = 113)** | | | | | | | | | | | | |
| All invasive fungal infection | 60 [51–69] | 60 [50–69] | 60 | 69 [55–80] | 67 [53–80] | 68 | 58 [39–75] | 76 [55–91] | 66 | 88 [47–99] | 37 [16–62] | 52 |
| Invasive aspergillosis | 42 [20–67] | 32 [15–54] | 36 | 43 [18–71] | 38 [15–65] | 40 | 75 [19–99] | 33 [7.49–70] | 46 | - | - | - |
| Invasive candidiasis | 83 [61–95] | 70 [58–81] | 76 | 67 [35–90] | 33 [16–55] | 44 | 100 [16–100] | 29 [3.67–71] | 45 | 100 [54–100] | 35 [14–62] | 52 |
| Other/ unspecified IFI | 20 [5.05–72] | 2.86 [0.72–15] | 5.00 | 0 [0–0] | 0 [0–0] | 0 | 50 [13–99] | 10 [0.25–45] | 17 | - | - | - |
| **Proven EORTC/MSG (N = 30)** | | | | | | | | | | | | |
| All invasive fungal infection | 77 [58–90] | 34 [23–46] | 47 | 71 [44–90] | 30 [17–47] | 42 | 86 [42–99] | 32 [13–57] | 47 | 100 [40–100] | 57 [18–90] | 73 |
| **Probable EORTC/MSG (N = 26)** | | | | | | | | | | | | |
| All invasive fungal infection | 54 [33–73] | 21 [11–32] | 30 | 60 [32–84] | 23 [11–38] | 33 | 78 [40–97] | 37 [16–62] | 50 | - | - | - |
| **Possible EORTC/MSG (N = 46)** | | | | | | | | | | | | |
| All invasive fungal infection | 61 [45–75] | 41 [29–54] | 49 | 78 [56–93] | 45 [29–62] | 57 | 40 [16–68] | 32 [13–57] | 36 | 100 [16–100] | 29 [3.67–71] | 45 |

Abbreviations: CI, confidence interval; EORTC/MSG, European Organization for Research and Treatment of Cancer/Invasive Fungal Infections Cooperative Group and the National Institute of Allergy and Infectious Diseases Mycoses Study Group; IFI, invasive fungal infection.

See S6 Table for performance classification statistics for the 'modified possible' and 'treated for IFI' classification.

ICD-10-AM codes B44.2, B44.8 and B44.9 denoting invasive aspergillosis, codes B37.5 and B37.6 denoting invasive candidiasis, and codes B48.0-B48.7 and B49 denoting other/unspecified IFI returned $F_1$ scores of 0%, yielding no improvement to the sensitivity and PPV (Table 4). The overall classification performance of the ICD-10-AM codes improved with larger sets of combined codes. Assignment of codes B44.0, B44.1 or B44.7, and B37.7 or B37.88, as a combined union set of codes, yielded the highest sensitivity, PPV and $F_1$ scores for invasive aspergillosis and invasive candidiasis, respectively (Tables 3 and 4). Only code B48.8 returned a sensitivity, PPV and $F_1$ estimate for the other/unspecific IFI category (Table 4).

The sensitivity of B44.x codes denoting invasive aspergillosis decreased for r-TERIFIC patients in receipt of 2 codes (Pr($B44.0 \cap B44.1$) = 5.26% [95% CI: 0.13–26]; Pr($B44.0 \cap B44.7$) = 0%; Pr($B44.1 \cap B44.7$) = 0%) compared to assignment of ≥1 code (42% [95% CI: 20–67], Table 3). Likewise, the sensitivity of B37.x codes denoting invasive candidiasis decreased for r-TERIFIC patients in receipt of 2 codes (Pr($B37.7 \cap B37.88$) = 17% [95% CI: 4.95–39]) compared to assignment of ≥1 code (83% [95% CI: 61–95], Table 3).

## Misclassification rate

Misclassification was greatest in patients with invasive aspergillosis coded as invasive candidiasis (n = 2; 13%). Of the 21 patients with invasive candidiasis, 2 (9.52%) were coded as invasive aspergillosis.

**Table 4. Performance (in percent) of different coding abstractions stratified by invasive fungal infection in the study cohort, N = 1,671.**

| ICD-10-AM code set(s) | Sensitivity [95% CI] | Positive predictive value [95% CI] | $F_1$ |
|---|---|---|---|
| **Invasive aspergillosis** | | | |
| *n = 1 code* | | | |
| B44.0 | 11 [1.30–33] | 50 [6.76–93] | 18 |
| B44.1 | 32 [13–57] | 38 [15–65] | 35 |
| B44.2 | 0 [0–0] | 0 [0–0] | 0 |
| B44.7 | 5.26 [0.01–26] | 50 [1.26–99] | 9.52 |
| B44.8 | 0 [0–0] | 0 [0–0] | 0 |
| B44.9 | 0 [0–0] | 0 [0–0] | 0 |
| *n≤2 codes* | | | |
| B44.0 ∪ B44.1 | 37 [16–62] | 37 [16–62] | 37 |
| B44.0 ∪ B44.7 | 16 [3.38–40] | 50 [12–88] | 24 |
| B44.1 ∪ B44.7 | 37 [16–62] | 39 [17–64] | 38 |
| *n≤3 codes* | | | |
| B44.0 ∪ B44.1 ∪ B44.7 | 42 [20–67] | 32 [15–54] | 36 |
| **Invasive candidiasis** | | | |
| *n = 1 code* | | | |
| B37.5 | 0 [0–0] | 0 [0–0] | 0 |
| B37.6 | 0 [0–0] | 0 [0–0] | 0 |
| B37.7 | 43 [22–66] | 100 [69–100] | 60 |
| B37.88 | 53 [30–74] | 59 [36–79] | 56 |
| *n≤2 codes* | | | |
| B37.7 ∪ B37.88 | 83 [61–95] | 70 [58–81] | 76 |
| **Other/unspecified invasive fungal infection** | | | |
| *n = 1 code* | | | |
| B48.0 | 0 [0–0] | 0 [0–0] | 0 |
| B48.1 | 0 [0–0] | 0 [0–0] | 0 |
| B48.2 | 0 [0–0] | 0 [0–0] | 0 |
| B48.3 | 0 [0–0] | 0 [0–0] | 0 |
| B48.4 | 0 [0–0] | 0 [0–0] | 0 |
| B48.7 | 0 [0–0] | 0 [0–0] | 0 |
| B48.8 | 20 [5.05–72] | 2.86 [0.72–15] | 5.00 |
| B49 | 0 [0–0] | 0 [0–0] | 0 |

Abbreviations: CI, confidence interval; ICD-10-AM, *International Statistical Classification of Diseases and Related Health Problems, Tenth Revision, Australian Modification*; ∪, union.

See S3 Table for definitions of each ICD-10-AM code.

## Discussion

This study is the first to describe the performance of administrative coding data to detect IFI in immunocompromised children with cancer. Overall sensitivity and PPV of ICD-10-AM codes for detection of clinically-confirmed IFI were moderate. However, sensitivity was improved for ascertainment of proven and possible IFI cases, in particular for invasive candidiasis, suggesting there is potential merit in using administrative coding data to signal medical record review for these discrete IFIs.

We found that ICD-10-AM codes alone were not sufficient to accurately classify IFI cases. In keeping with earlier estimates reported in Chang *et al.* [12], we observed an overall

sensitivity and PPV of 60%, indicating that administrative coding data alone are not sufficient to reliably detect true cases of IFI in paediatric patients. The performance of coding data for IFI case detection was enhanced when subsets of proven IFI were examined, suggesting that where confirmatory laboratory results are available, then the quality of coding may be improved. Accuracy and completeness of medical record documentation likely contributes to this variation with one study showing that 97% of fungaemia cases were assigned an IFI code when fungaemia was explicitly documented in the medical record, as opposed to only 42% of cases when only microbiology results were used [24]. While underlying malignancy is important for evaluating IFI risk [4], our findings suggest that cancer diagnosis is less relevant to understanding the classification performance of administrative coding data for IFI.

In addition, we observed a difference in the performance of coding data for accurate detection of specific subsets of fungal infection. Cases of invasive candidiasis were more accurately coded than invasive aspergillosis cases. We propose that this may be related to readily available and simple diagnostic tests for yeast infections, in comparison to heterogenous diagnostic testing and the requirement for interpretation of imaging and laboratory results in order to identify mould infections. These factors could impact upon coding practices, particularly where microbiology, histology and radiological findings require integration by clinicians, with documentation in medical files, to facilitate accurate coding by clinical coders.

Cases of invasive aspergillosis were most frequently misclassified as invasive candidiasis in the coded data. Although the number of invasive aspergillosis cases in the gold-standard data were small (N = 15), our findings are likely indicative of the uncertainty in discriminating between yeast and mould infections at the clinical coding level. A recent qualitative study [25] identified clinical coders' experience and awareness of IFI as a factor associated with discordant coding. Although it is a reasonable assumption that clinical coder experience is associated with our misclassification estimates, in the setting of IFI where clinical case definitions are complex, it is conceivable that other factors are at play. This includes the complexity of translating clinical data indicating invasive aspergillosis into ICD-10-AM [24, 26], the absence of clear definitions [27, 28], subjective interpretation of existing guidelines [24, 25, 27], delays in diagnosis [29], and the review of multiple data sources to make a confirmatory diagnosis of mould infection [1, 2, 30]. This setting underscores the importance of clear, complete, legible and standardised documentation of IFI to mitigate misclassification in current coding workflows.

We noted variation in classification performance according to specific code sets. Our results indicate that algorithms including the largest combination of specific IFI code sets yield the highest probability for case ascertainment in hospitalised paediatraic cancer patients. Notwithstanding, the fact that the $F_1$ score for specific invasive aspergillosis code abstractions (B44.0 ∪ B44.1 ∪ B44.7) is still low-to-moderate ($F_1$ = 36%, Tables 3 and 4) underscores that although these specific codes are the most sensitive starting point to signal medical chart review, existing coding rules are an unreliable indicator for invasive mould infections when used in isolation. Importantly, the sensitivity of ICD-10-AM decreases from 42% to 5.26% and 83% to 17% when comparing patients assigned one versus two codes denoting invasive aspergillosis and invasive candidiasis, respectively. Mathematically, the subset of true positive cases (numerator) diminishes as the number of assigned IFI codes increases (and the case definition therefore becomes more specific), whilst the number of gold-standard IFI cases (denominator) remains fixed. For example, true positives with 3 IFI codes is a subset of true positives with ≥2 IFI codes, which is a subset of true positives with ≥1 IFI code. Alternatively, {patients with 3 codes} ⊆ {≥2 codes} ⊆ {≥1 code}.

Methodological differences between the Australian Coding Standards and EORTC/MSG definitions likely contribute to our moderate overall $F_1$ score of 60%. Clinical coders must

adhere to rigid coding rules in accordance with Australian Coding Standards in the same way that clinicians adhere to complex and comprehensive criteria for IFI (i.e. EORTC/MSG), but these two sets of criteria may not directly match. This disconnect in clinical case definitions is a fundamental drawback in using ICD-10 codes as a reproducible proxy for IFI given cases detected according to clinical criteria may not reflect coded cases using ICD-10-AM. For example, clinical coders' reliance on microbiology and histology records to identify cases of IFI in line with current coding rules can be subject to ascertainment bias in the coded data, given many IFIs are diagnosed according to a combination of metrics, namely clinical acumen, radiological findings and serum antigen testing [12, 26]. Notwithstanding, strategic imperatives to mitigate erroneous coding of IFI are likely two-fold. First, harmonisation of clinical EORTC/MSG definitions with existing Australian Coding Standards may help safeguard accurate detection of IFI in the coded data by reducing ascertainment of false positive cases (for example, our high number of false positive cases [N = 38] coded as 'candidiasis of other sites' [code B37.88]). In fact, recent qualitative research proposes the use of Systematized Nomenclature of Medicine–Clinical Terms (SNOMED-CT) codes in electronic health records as a more granular tool to standardise terminology and facilitate clinical coding of complex diseases [25, 31, 32]. Second, ensuring that chart documentation is complete, legible and streamlined will ensure clinical coders have sufficient access to the data required to assign the appropriate IFI code(s) [25, 33, 34].

Our high classification estimates for invasive candidiasis suggest that administrative coding data may be a feasible proxy to facilitate existing surveillance methods of yeast infection. Owing to the availability and easier interpretation of confirmatory diagnostic data indicating invasive candidiasis compared to invasive aspergillosis [35], the sensitivity and PPV of the administrative coding data are high ($F_1$ = 76%). These findings substantiate potential merit in its use as a signal to trigger medical record review. Current surveillance of IFI is manual, onerous, time-consuming and resource-intensive [1, 2, 4, 12, 24]. However, use of ICD-10 codes as a feasibly available surrogate measure for invasive candidiasis may help restrict medical chart reviews to patients most likely presenting with yeast infection, therefore mitigating unnecessary record review. Our promising classification results also suggest there may be value in using ICD-10-AM codes for population-based monitoring of invasive candidiasis (codes B37.x) in paediatric populations.

Limitations of the current study include the fact that single-centre experience was evaluated, and findings may not reflect clinical coding performance and differences in other paediatric haematology-oncology units [36, 37]. Second, ICD-10 is an amalgamation of diagnostic information into a codified, monohierarchical, medical lexicon which does not discriminate between EORTC/MSG classifications, therefore rendering the data insufficient for fungal surveillance based on classification of proven/probable/possible IFI. Third, the wide 95% confidence intervals for our classification estimates (Table 3) are attributed to a small sample size of true positive cases stratified by type of IFI and EORTC/MSG criteria. Further, although the one-month average time lag [38] for hospital diagnoses to be coded make ICD-10-AM unsatisfactory for real-time IFI surveillance, our data indicate potential merit in using invasive candidiasis codes (B37.x) to signal retrospective detection of potentially missed cases.

## Conclusions

In conclusion, we demonstrate moderate performance of ICD-10-AM codes for detection of IFI in children with cancer. Coding of invasive fungal infections having greater diagnostic certainty according to EORTC/MSG criteria (i.e. proven IFI), as well as yeast infections, resulted in higher sensitivity for case ascertainment. Findings suggest that while administrative coding

data are not an accurate reflection of overall IFI disease burden, these data may provide an acceptable reflection of relative disease burden and signal a medical chart review for specific IFI categories (namely, proven/possible IFI and yeast infections) in paediatric patients with cancer. Future studies are required to assess the utility of ICD-10-AM data for these specific infections to detect changes in disease burden and longitudinally monitor quality improvement activities.

## Supporting information

**S1 Table. STROBE statement—checklist of items that should be included in reports of cohort studies.**
(PDF)

**S2 Table. The RECORD statement—checklist of items, extended from the STROBE statement, that should be reported in observational studies using routinely collected health data.**
(PDF)

**S3 Table. ICD-10-AM coding conventions for invasive fungal infection, haematological malignancy and solid tumour neoplasms.**
(PDF)

**S4 Table. Australian Classification for Health Intervention codes denoting allogeneic and autologous haematopoietic stem cell transplantations.**
(PDF)

**S5 Table. The STARD 2015 statement—checklist of essential items for reporting diagnostic accuracy studies.**
(PDF)

**S6 Table. Performance (in percent) of administrative coding data for case detection of 'modified possible' and 'treated for invasive fungal infection' classifications.**
(PDF)

## Acknowledgments

The authors thank Ms. Belinda Zambello of the Victorian Paediatric Integrated Cancer Service for preparing the administrative coding data from the Victorian Admitted Episodes Dataset for collection.

## Author Contributions

**Conceptualization:** Jake C. Valentine, Leon J. Worth, Gabrielle M. Haeusler.

**Data curation:** Jake C. Valentine, Julia E. Clark, Gabrielle M. Haeusler.

**Formal analysis:** Jake C. Valentine.

**Funding acquisition:** Jake C. Valentine, Gabrielle M. Haeusler.

**Investigation:** Jake C. Valentine, Leon J. Worth, Gabrielle M. Haeusler.

**Methodology:** Jake C. Valentine, Leon J. Worth, Karin M. Verspoor, Gabrielle M. Haeusler.

**Project administration:** Jake C. Valentine, Gabrielle M. Haeusler.

**Resources:** Jake C. Valentine.

**Software:** Jake C. Valentine.

**Supervision:** Leon J. Worth, Karin M. Verspoor, Lisa Hall, Gabrielle M. Haeusler.

**Validation:** Jake C. Valentine, Gabrielle M. Haeusler.

**Visualization:** Jake C. Valentine, Karin M. Verspoor, Gabrielle M. Haeusler.

**Writing – original draft:** Jake C. Valentine.

**Writing – review & editing:** Jake C. Valentine, Leon J. Worth, Karin M. Verspoor, Lisa Hall, Daniel K. Yeoh, Karin A. Thursky, Julia E. Clark, Gabrielle M. Haeusler.

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
