## [Decision Letter · Decision Letter 0]

24 Jul 2020

PONE-D-20-16766

Classification performance of administrative coding data for detection of invasive fungal infection in paediatric cancer patients

PLOS ONE

Dear Dr. Valentine,

Thank you for submitting your manuscript to PLOS ONE. After careful consideration, we feel that it has merit but does not fully meet PLOS ONE’s publication criteria as it currently stands. Therefore, we invite you to submit a revised version of the manuscript that addresses the points raised during the review process.

We look forward to receiving your revised manuscript.

Kind regards,

Ales Vicha, M.D., PhD

Academic Editor

PLOS ONE

Journal Requirements:

2.We note that you have indicated that data from this study are available upon request. PLOS only allows data to be available upon request if there are legal or ethical restrictions on sharing data publicly. For information on unacceptable data access restrictions, please see http://journals.plos.org/plosone/s/data-availability#loc-unacceptable-data-access-restrictions.

3.Thank you for stating the following in the Competing Interests section:

[G.M.H. and J.E.C. report investigator-initiated grant funding from Gilead Sciences. K.M.V. reports grant funding from the National Health and Medical Research Council. All other authors declare no conflicts of interest relevant to this article.].

Additional Editor Comments (if provided):

The text of the manuscript is well written, but it needs minor adjustments. It is necessary to adjust the text according to the comments of reviewers.

Reviewers' comments:

Reviewer's Responses to Questions

**Comments to the Author**

1. Is the manuscript technically sound, and do the data support the conclusions?

Reviewer #1: Yes

Reviewer #2: Yes

2. Has the statistical analysis been performed appropriately and rigorously? 

Reviewer #1: Yes

Reviewer #2: Yes

3. Have the authors made all data underlying the findings in their manuscript fully available?

Reviewer #1: Yes

Reviewer #2: Yes

4. Is the manuscript presented in an intelligible fashion and written in standard English?

Reviewer #1: Yes

Reviewer #2: Yes

5. Review Comments to the Author

Reviewer #1: This is a very interesting manuscript and it is well written and study well conducted. My only question is if the authors have tried to analyze the PPV by stratifying the number of codes assigned. In other words: is there any difference in sensitivity of the diagnostic code if the patient received 1 only code vs 2 or more? In the manuscript the authors stated >=1 but it would be interesting to see if any difference.

Reviewer #2: The authors evaluated the classification performance of ICD-10-AM codes for detecting IFI using a gold-standard dataset of confirmed IFIs in paediatric cancer patients at a quaternary referral centre. But there were some limitations and unclear descriptions that they need to revise.

Minor

1. Page 8, line 180, “ICD-10-AM codes: C00.x - C76.x, C80.x, C81.x [0/1] - C88.x [0/1], C90.x [0/1] - C96.x [0/1], and D46”

In table 3, some of code were seemed to not be included above descriptions. Please check it.

2. Table 2 the number described in parenthesis ( )

Please describe the number that described in parenthesis ( ). “Male 55” meant percentage of male? Please describe the meaning of other numbers, too.

3. Although the results of EORTC/MSG were described in Table 3, no description exist in the text. Please add the results of EORTC/MSG in the results section.

4. Were there the case with two or more infection of fungi? If the two or more infection of fungi exist, how did the authors manage these cases?

6. PLOS authors have the option to publish the peer review history of their article (what does this mean?). If published, this will include your full peer review and any attached files.

Reviewer #1: No

Reviewer #2: **Yes: **Toshihiko Mayumi

---

## [Author Response · Author response to Decision Letter 0]

1 Aug 2020

Dr. Ales Vicha, M.D., Ph.D.

Academic Editor, 

PLOS ONE

1st August 2020

Dear Dr. Vicha,

R.E.: Article ID: PONE-D-20-16766

Please find attached revisions to the attached manuscript, entitled ‘Classification performance of administrative coding data for detection of invasive fungal infection in paediatric cancer patients’.

All reviewer comments have been considered, with modifications made to the manuscript. Please find enclosed both a clean version as well as the copy containing the tracked changes to the manuscript. 

Please note that there are ethical restrictions concerning public provision of these data due to the sensitive nature of medical information contained in these datasets. As previously stipulated in our Data Availability Statement, data sharing is restricted to the researchers, clinical staff and the individual patient’s healthcare provider participating in the project analysis. The ethics approval (HREC/59636/RCHM-2019) maintains that public data provision is contingent on authorisation from the Principal Investigator and associated ethics applications to the Royal Children’s Hospital Human Research Ethics Committee for researchers who meet the criteria for access to confidential data. Contact details for the relevant ethics committee to which data requests may be sent is rch.ethics@rch.org.au.

Updated Financial Disclosure Statement:

J.C.V. was supported by an Australian Government Research Training Program Scholarship (grant number: 290465) awarded by the University of Melbourne (URL: https://www.education.gov.au/research-training-program) and a Cardinal Health Infection Control Scholarship awarded by the Australasian College for Infection Prevention and Control (URL: https://www.cardinalhealth.com/en.html). The TERIFIC study was supported by an Investigator Initiated Grant (grant number: IN-AU-131-1314) from Gilead Sciences, Inc. (https://www.gilead.com/). The funders had no role in study design, data collection and analysis, decision to publish, preparation or review of the manuscript.

Updated Competing Interest Statement:

I have read the journal’s policy and the authors of this manuscript have the following competing interests: G.M.H. and J.E.C. report investigator-initiated grant funding from Gilead Sciences, Inc. K.M.V. reports grant funding from the National Health and Medical Research Council. All other authors declare no conflicts of interest relevant to this article. This does not alter our adherence to PLOS ONE policies on sharing data and materials.

Please also note that the authors have updated the visual aesthetics to Figure 1. All content in Figure 1 is identical to the original submission. Figure 1 has been uploaded to PACE and satisfies PACE requirements. We look forward to hearing from you regarding publication of this manuscript.

Yours sincerely,

Mr. Jake C. Valentine, BBiomedSc(Hons)

(corresponding author) 

T: (03) 8559 7036 | F: (03) 8559 7999 | E: jvalentine@student.unimelb.edu.au or Jake.Valentine@petermac.org

A: 305 Grattan Street, Melbourne, Victoria, Australia 3000

Response to reviewers: PONE-D-20-16766

Reviewer 1

Comment 1

This is a very interesting manuscript and it is well written and study well conducted. My only question is if the authors have tried to analyze the PPV by stratifying the number of codes assigned. In other words: is there any difference in sensitivity of the diagnostic code if the patient received 1 only code vs 2 or more? In the manuscript the authors stated >=1 but it would be interesting to see if any difference.

Response:

The authors thank the reviewer for their comment and agree that determining any variation in sensitivity and PPV of the coding data according to the number of assigned codes in the r-TERIFIC gold-standard dataset is highly relevant to the current study. Mathematically, the sensitivity decreases for r-TERIFIC patients in receipt of ≥2 codes within each combination of IFI codes, as the number of true positive cases naturally diminishes (namely, the set of invasive aspergillosis patients with 3 codes is a subset of patients with 2 codes, which is a subset of patients with 1 code etc). Please see page 18 (lines 392-397) and page 23 (lines 485-492) for the results and discussion on this point. Notwithstanding, the authors acknowledge that variation in the classification performance of specific codes within each combination of IFI codes is likely, given our current methodology evaluates the performance of a composite set of codes stratified by IFI type (according to composite coding definitions in Table S3). Thus, we have expanded our analysis to also evaluate any variation in the sensitivity, PPV and F1 scores for each specific code within each IFI code combination, first as a single set of codes, then as increasing sets of codes (Table 4, methodology discussed on pages 9 and 10, lines 227-239). See Table 4 and page 18 (lines 382-390) for results). This analysis will give an indication as to the individual sets of IFI codes yielding the highest sensitivity, PPV and F1 estimates in development of a coding algorithm for case detection of IFI in this population.

Reviewer 2

Comment 1

The authors evaluated the classification performance of ICD-10-AM codes for detecting IFI using a gold-standard dataset of confirmed IFIs in paediatric cancer patients at a quaternary referral centre. But there were some limitations and unclear descriptions that they need to revise.

Page 8, line 180, “ICD-10-AM codes: C00.x - C76.x, C80.x, C81.x [0/1] - C88.x [0/1], C90.x [0/1] - C96.x [0/1], and D46”

In table 3, some of code were seemed to not be included above descriptions. Please check it.

Response:

The authors thank the reviewer for their comment. The authors have revisited the ICD-10-AM coding definitions for the malignancy subgroups reported in Table 3 and can confirm that all coding rules for each cancer type are correctly specified and included as stipulated in Table S3 (Supporting Information). The IFI coding performance in any patient in receipt of a Principal Diagnosis code (Australian Coding Standards 0001) listed on page 8, lines 185-186, were included in the ‘All cancers’ column of the analysis in Table 3. Given a significant majority of our cohort were diagnosed with acute lymphoblastic leukaemia, acute myeloid leukaemia and neuroblastoma (as indicated in Table 2), we conducted a subgroup analysis of IFI coding performance in these discrete subpopulations in order to discern any variation in IFI coding sensitivity based on underlying malignancy. 

Comment 2

Table 2 the number described in parenthesis ( )

Please describe the number that described in parenthesis ( ). “Male 55” meant percentage of male? Please describe the meaning of other numbers, too.

Response:

Numbers in the parenthesis denote percentages as indicated in the Table column heading. To avoid confusion regarding statistics in squared brackets, the population size (i.e. N=1,671) has been moved from the Table column heading to the Table title. Therefore, numbers in squared brackets represent the descriptive statistics indicated in the respective Table rows. See Table 2, page 14.

Comment 3

Although the results of EORTC/MSG were described in Table 3, no description exist in the text. Please add the results of EORTC/MSG in the results section.

Response:

The authors thank the reviewer for their comment. The sensitivity, PPV and F1 scores of the IFI codes in the studied population after stratifying by EORTC/MSG classification have now been inserted in the Results Section (page 15, lines 364-368).

Comment 4

Were there the case with two or more infection of fungi? If the two or more infection of fungi exist, how did the authors manage these cases?

Response:

Regarding use of ≥2 discrete IFI codes assigned to the same patient: we have indicated in the Methods Section that duplicate IFI codes denoting the same IFI in the same hospitalisation, as well as those reported at the time of admission in subsequent hospitalisations, were considered the same IFI and were counted only once per patient. Multiple discrete IFI codes appearing in the same hospitalisation per patient were counted as separate IFI episodes (page 8, lines 193-196). Regarding cases with ≥2 discrete gold-standard IFI diagnoses: patients with ≥2 mutually exclusive gold-standard IFI diagnoses in the r-TERIFIC dataset were counted as individual gold-standard cases (for example, one patient with invasive aspergillosis and invasive candidiasis was counted as one case of invasive aspergillosis and one case of invasive candidiasis accordingly). This occurred in only 1 instance. This clarification has now been inserted in the Methods Section (page 8, lines 196-200).

---

## [Decision Letter · Decision Letter 1]

26 Aug 2020

Classification performance of administrative coding data for detection of invasive fungal infection in paediatric cancer patients

PONE-D-20-16766R1

Dear Dr. Valentine,

We’re pleased to inform you that your manuscript has been judged scientifically suitable for publication and will be formally accepted for publication once it meets all outstanding technical requirements.

Kind regards,

Ales Vicha, M.D., PhD

Academic Editor

PLOS ONE

Additional Editor Comments (optional):

Reviewers' comments:

Reviewer's Responses to Questions

**Comments to the Author**

1. If the authors have adequately addressed your comments raised in a previous round of review and you feel that this manuscript is now acceptable for publication, you may indicate that here to bypass the “Comments to the Author” section, enter your conflict of interest statement in the “Confidential to Editor” section, and submit your "Accept" recommendation.

Reviewer #1: All comments have been addressed

Reviewer #2: All comments have been addressed

2. Is the manuscript technically sound, and do the data support the conclusions?

Reviewer #1: Yes

Reviewer #2: Yes

3. Has the statistical analysis been performed appropriately and rigorously? 

Reviewer #1: Yes

Reviewer #2: Yes

4. Have the authors made all data underlying the findings in their manuscript fully available?

Reviewer #1: Yes

Reviewer #2: Yes

5. Is the manuscript presented in an intelligible fashion and written in standard English?

Reviewer #1: Yes

Reviewer #2: Yes

6. Review Comments to the Author

Reviewer #1: Thank you for the clarifications. The comments were addressed properly, this reviewer does not have additional edits.

Reviewer #2: The authors evaluated the classification performance of ICD-10-AM codes for detecting IFI using a gold-standard dataset of confirmed IFIs in paediatric cancer patients at a quaternary referral centre. After the revisions according to the reviewers, the quality of the paper had improved.

7. PLOS authors have the option to publish the peer review history of their article (what does this mean?). If published, this will include your full peer review and any attached files.

Reviewer #1: No

Reviewer #2: **Yes: **Toshihiko Mayumi, M.D. & Ph.D.

---

## [Editor Report · Acceptance letter]

31 Aug 2020

PONE-D-20-16766R1 

Classification performance of administrative coding data for detection of invasive fungal infection in paediatric cancer patients 

Dear Dr. Valentine:

I'm pleased to inform you that your manuscript has been deemed suitable for publication in PLOS ONE. Congratulations! Your manuscript is now with our production department. 

Kind regards, 

on behalf of

Dr. Ales Vicha 

Academic Editor

PLOS ONE